# Autonomous platform for solution processing of electronic polymers

Chengshi Wang [1,7], Yeon-Ju Kim [1,7], Aikaterini Vriza [1,7], Rohit Batra [1,5,7], Arun Baskaran [1,6], Naisong Shan [2], Nan Li [2], Pierre Darancet [1], Logan Ward [3], Yuzi Liu [1], Maria K. Y. Chan [1], Subramanian K.R.S. Sankaranarayanan [1,4], H. Christopher Fry [1], C. Suzanne Miller [1], Henry Chan [1] ✉ & Jie Xu [1,2] ✉

The manipulation of electronic polymers' solid-state properties through processing is crucial in electronics and energy research. Yet, efficiently processing electronic polymer solutions into thin films with specific properties remains a formidable challenge. We introduce Polybot, an artificial intelligence (AI) driven automated material laboratory designed to autonomously explore processing pathways for achieving high-conductivity, low-defect electronic polymers films. Leveraging importance-guided Bayesian optimization, Polybot efficiently navigates a complex 7-dimensional processing space. In particular, the automated workflow and algorithms effectively explore the search space, mitigate biases, employ statistical methods to ensure data repeatability, and concurrently optimize multiple objectives with precision. The experimental campaign yields scale-up fabrication recipes, producing transparent conductive thin films with averaged conductivity exceeding 4500 S/cm. Feature importance analysis and morphological characterizations reveal key design factors. This work signifies a significant step towards transforming the manufacturing of electronic polymers, highlighting the potential of AI-driven automation in material science.

The control of solid-state properties through molecular assembly processes of electronically functional materials has been a decades-long pursuit in the electronics and energy industries. Electronic polymers, known for their unique electronic properties, mechanical softness, and low-cost production, have been studied extensively and utilized in printable electronics, wearable and bioelectronics, and energy devices[1–6]. Yet, the high-throughput processing of electronic polymer solutions into thin films with desirable properties remains a major challenge in device manufacturing. The processing of these electronic nanometer-thick films typically involves using various formulations under rapid flows and stressors (e.g., heating), which are

highly non-equilibrium conditions that can lead to unpredictable morphological variabilities. Therefore, achieving precise control over the morphology of electronic polymer thin films is crucial for realizing the desired functional properties and ensuring uniformity. However, the large number of parameters and their complex relationships in the processing of electronic polymers presents a major challenge in quickly achieving a desired performance goal, resulting in years of dedicated effort for designing and optimizing new electronic polymer materials that exhibit enhanced functionalities.

The solution manufacturing of electronic polymers into functional layers involves three main steps: solution formulation, thin film

[1]Nanoscience and Technology Division, Argonne National Laboratory, Lemont, IL, USA. [2]Pritzker School of Molecular Engineering, The University of Chicago, Chicago, IL, USA. [3]Data Science and Learning Division, Argonne National Laboratory, Lemont, IL, USA. [4]Department of Mechanical and Industrial Engineering, University of Illinois, Chicago, IL, USA. [5]Present address: Department of Metallurgical and Materials Engineering, Indian Institute of Technology Madras, Chennai, India. [6]Present address: Corning Incorporated, Corning, NY, USA. [7]These authors contributed equally: Chengshi Wang, Yeon-Ju Kim, Aikaterini Vriza, Rohit Batra. ✉e-mail: hchan@anl.gov; xuj@anl.gov

coating on a substrate, and post-processing. Despite decades of experimentation[7–14], our understanding and control of thin film processing mechanisms remains limited due to our reliance on heuristics and human scientists in establishing comprehensive, unbiased datasets[7,15]. Efficiently collecting these datasets is crucial for uncovering the intricate, high-dimensional relationships between formulation, processing, and material properties. Recent advancements in automated robotic technologies have significantly increased productivity in medical and materials science research fields by offloading repetitive work from human scientists[16–20]. Coupling this with the advent of machine learning (ML) for data analysis and artificial intelligence (AI) as a cognitive assistant for navigating complex parameter spaces has inspired the development of modern autonomous laboratories, also known as self-driving laboratories[21]. These laboratories complement combinatorial experiments and have achieved significant progress in expediting the optimization and discovery of various materials, yet creating one for solution manufacturing of electronic polymer films faces challenges[22–30]. The inherent complexities in polymer processing-property relationships often lead to reduced experimental throughput and small datasets with high experimental uncertainties, which limits the effective utilization of AI/ML for exploring the multi-dimensional space associated with the processing, structure, and properties of polymers. Hence, existing AI-guided thin film processing studies are predominantly restricted to a small set of experimental parameters, e.g., pre-syn[28,31] and a single material property[28,31–34], while in practice, real-world applications necessitate the simultaneous consideration of many experimental parameters and multiple material properties. To address these challenges, it is essential to put emphasis on the quality and repeatability of experimental data and leverage learning algorithms that are robust to small datasets.

In this work, we introduce an automated solution processing platform implemented within Polybot[35]—a state-of-the-art self-driving laboratory. This platform enables efficient exploration of a multi-dimensional parameter space encompassing the formulation, coating, and post-processing of electronic polymer thin films. Using importance-guided Bayesian optimization, a tailored learning algorithm that handles multiple objectives, Polybot strategically explored undersampled regions of the search space and exploited available data to produce thin films with high-conductivity and low defects[28,31–34]. In addition, we implemented statistical data analysis methods to ensure experimental repeatability, a foundation to quality datasets and accurate AI/ML predictions. Our results demonstrated a successful autonomous experimental campaign and help design recipes for scale-up fabrication of transparent conductive thin films that achieved an averaged conductivity of over 4500 S/cm. Furthermore, the data revealed important factors influencing the defects and conductivity of electronic polymers, which are supported by in-depth characterizations of the solution-state structures and solid-state morphologies. Polybot represents an ongoing effort to enhance our understanding of electronic polymer thin films, and its continuous development aims to advance the field by pushing the boundaries of materials discovery.

## Results

### Automated solution processing of electronic films

In this study, we utilized a robot-operated experimental workflow for exploring the conditions of solution-processed electronic polymer thin films towards desired film properties (Fig. 1a). The automated platform is equipped with liquid/substrate/vial handling stations, a solution mixing station, blade-coating station, blade cleaning station, annealing station, as well as a range of online characterization and analytics systems, encompassing imagining and thickness characterization modules, along with an automated probe station connected to an electrical characterization system (Fig. 1b). The automated platform can complete an entire experimental loop—formulation, processing, post-processing, and conductivity measurement—in approximately

15 min per sample, enabling a throughput of around 100 samples per day with great repeatability. The Polybot control software orchestrates the experimental workflows, data flow, and ML-based automated data/performance analysis (Supplementary Movie 1).

Poly(3,4-ethylenedioxythiophene) doped with poly(4-styrenesulfonate) (PEDOT:PSS) is chosen as an exemplary material in this study (see Methods) to showcase the autonomous experimentation methodology and highlight our innovation. Despite PEDOT:PSS being acknowledged as a highly conductive polymer, its conductivity and coating defects (e.g., dewetted regions, holes) are notably sensitive to formulation and processing conditions[36]. Our strategies for achieving highly conductive PEDOT:PSS films are grouped into three main categories: (1) incorporating additives to improve connectivity between PEDOT-rich domains, facilitating high charge carrier mobility, (2) employing directional film coating methods to introduce morphological alignment, and (3) implementing solvent post-process treatments to enhance morphological ordering and/or remove PSS, which is insulating.

Diverging from traditional research methodologies that vary one parameter at a time while keeping others fixed, our experiments simultaneously vary all parameters with the guidance of probabilistic AI/ML and statistical analysis. Our framework optimizes properties of PEDOT:PSS thin films using multi-objective Bayesian optimization enhanced by probabilistic sampling (Fig. 1c). The properties of PEDOT:PSS thin films are intricately influenced by numerous thermodynamic and kinetic states during formulation and deposition processes, factors such as polymer chain conformation, aggregate types in solution, structure regulation/relaxation during coating, and subsequent structural development during post-treatments. Given the interconnected nature of these states, individual control becomes challenging. In our autonomous experiments, seven experimental parameters were concurrently adjusted to modulate the polymer solution-state structures, control assembly during coating, and manage structural regulation through post-treatment (Fig. 1d, Methods). These parameters encompassed additive types, additive ratios, blade-coating speeds, blade-coating temperatures, post-processing solvents, prost-processing coating speeds, and post-processing coating temperatures (Fig. 1e, Supplementary Tables 1 and 2, and Supplementary Fig. 1). This holistic approach allows us to discern the relative importance and specific influence of these factors in the quest for optimal manufacturing conditions of thin films with desired electronic and coating properties.

The primary objective of our experiments is to maximize the electrical conductivity of PEDOT:PSS thin films while achieving low coating defects. To accomplish this, Polybot leverages automated stations for formulating polymer solutions, coating the thin films, assessing their processability, and evaluating their electrical conductivity. The quantification of film processability is outlined in Fig. 2a and Supplementary section 1.1. The procedure estimates the uniformity of thin films using color (hue) information extracted from a set of top-view images captured by a camera (Supplementary Fig. 2). Close-up images of the substrate and thin film are extracted using image processing and computer vision techniques including thresholding, Harris corner detection, and perspective transformation, which corrects for minor optical aberrations and minimizes any subtle translational and rotational variants in the placement of the samples by the robot. The procedure for thin film electrical conductivity measurements is outlined in Fig. 2b and Supplementary section 1.2. Eight separate current-voltage (IV) curves are measured across different regions of the sample, using a 4-point collinear probe station connected to a Keithley 4200. The conductivity values are then calculated from resistivity extracted from the IV curves and normalized by film thicknesses that are measured in the specific local regions where the IV curves are obtained (Supplementary Figs. 3 and 4).

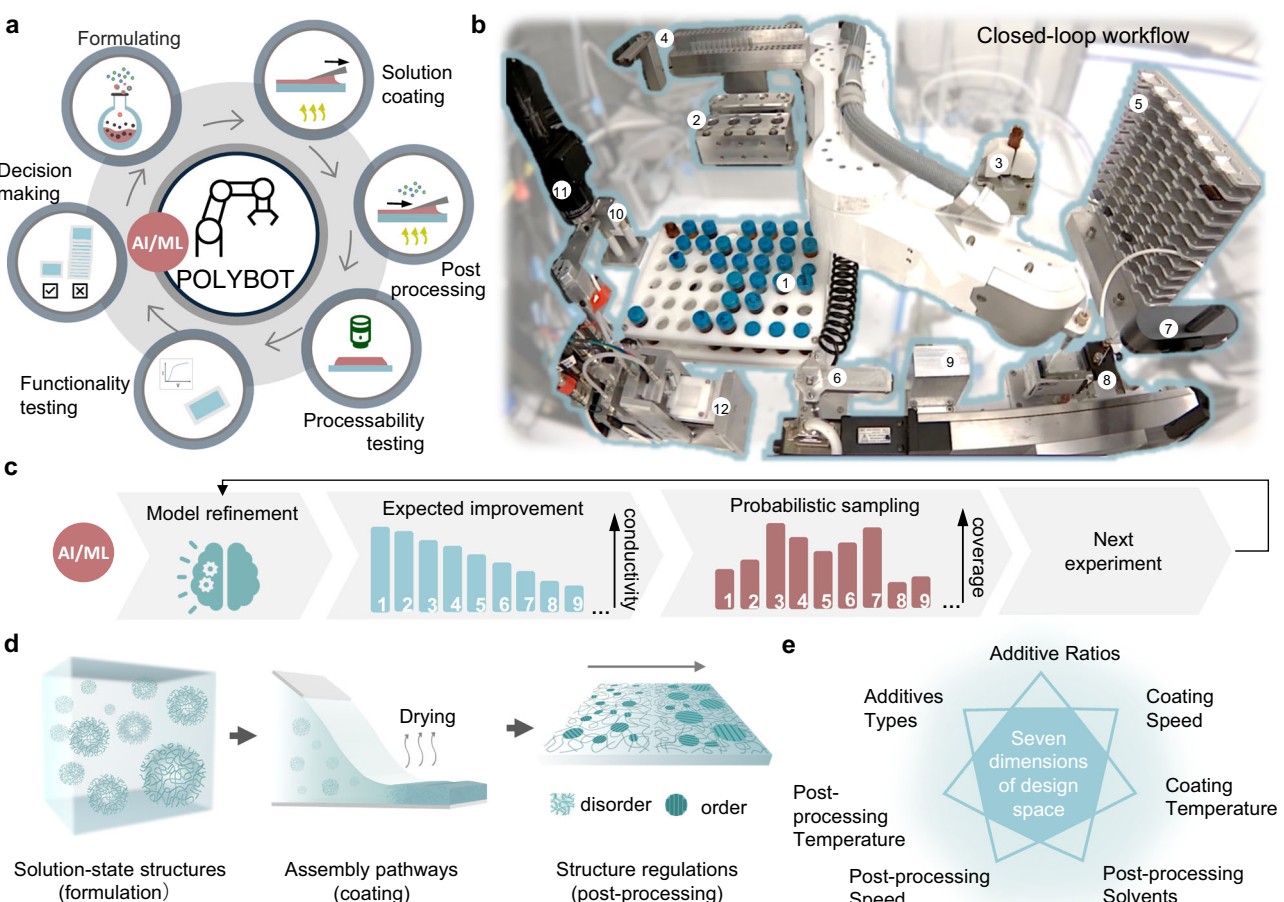

**Fig. 1 | A closed-loop electronic thin film discovery platform in self-driving laboratory Polybot. a** Schematic illustrating the consecutive steps in the autonomous experimental workflow. **b** Image of the modular automated platform which includes (1) solution storage rack, (2) solution heating and mixing module, (3) capping and uncapping system, (4) pipette rack, (5) substrate rack, (6) substrate gripper, (7) imaging station, (8) blade-coating station, (9) blade cleaning station, (10) annealing station, (11) thickness characterization station, (12) electrical characterization station. **c** Schematic of the iterative multi-objective optimization strategy based on advanced learning algorithms enhanced by probabilistic sampling, strategically exploring undersampled areas of the search space and exploited available data to produce thin films with superior processability and conductivity. **d** Complex assembly pathways of electronic polymers from solution to thin films. **e** The total searches space to optimize the conductivity of the PEDOT:PSS. For this seven-variable problem, the full design space has 933,120 distinct data points.

One of the major challenges in automated processing of PEDOT:PSS thin films is the high inherent uncertainty associated with their measured conductivity values, especially for films that are non-uniform due to poor film processability or dewetting. To ensure the repeatability of our experiments, Polybot performs at least two trials and up to four trials for every sample. A statistical analysis approach is implemented to eliminate invalid values and to determine the appropriate number of trials required (Fig. 2b). Specifically, the learning algorithm in Polybot only utilizes the two most statistically significant trials of each sample, which is determined through a normality check using the Shapiro–Wilk test[37] with a significance level of 0.03 and a two-sample $t$-test with a significance level of 0.005 (Supplementary section 1.3).

**From autonomous robotic experiments to scale-up fabrication**
Guided by AI/ML, our experiments concurrently adjust all parameters, elucidating the relative importance of experimental factors for achieving optimal manufacturing conditions of electronic thin films. In this investigation, the boundaries and increments of the seven experimental parameters (Supplementary Table 1) are set based on established conventions reported in relevant PEDOT:PSS thin film literatures, as well as the limitation, sensitivity, and tunability of our hardware modules[36]. Despite the discretization of continuous variables, the exploration encompasses a total of 933,120 possible

experimental conditions involving the formulation, coating, and post-processing of PEDOT:PSS thin films. Navigating this extensive search space without prior data is efficiently managed by Polybot, leveraging materials property prediction models and an importance-guided Bayesian Optimization (BO) approach to utilize existing data and explore undersampled processing conditions (Supplementary section 2).

At the start of our autonomous experiment, 30 conditions were uniformly sampled from the search space using the Latin Hypercube Sampling (LHS) method (Supplementary section 2.1). These data points coarsely cover a wide region of the search space and serve as initial training data for the prediction models: a Gaussian processes regression (GPR) model for electrical conductivity (Supplementary section 2.2) and a Gaussian kernel density estimation (KDE) model for film defects (Supplementary section 2.3). The GPR model predicts electrical conductivity of thin films based on all experimental parameters whereas the KDE model estimates a percentage of the thin film coverage area on the substrate prior to post-processing steps. This estimation is based on the top three important experimental parameters identified by Shapley feature importance analysis of the training data: DMSO concentration, blade-coating temperature, and blade-coating velocity (Supplementary section 3.2, Supplementary Fig. 8, and Supplementary Tables 4 and 5). Notably, the film coverage was reliably predicted using the train data alone, likely due to their

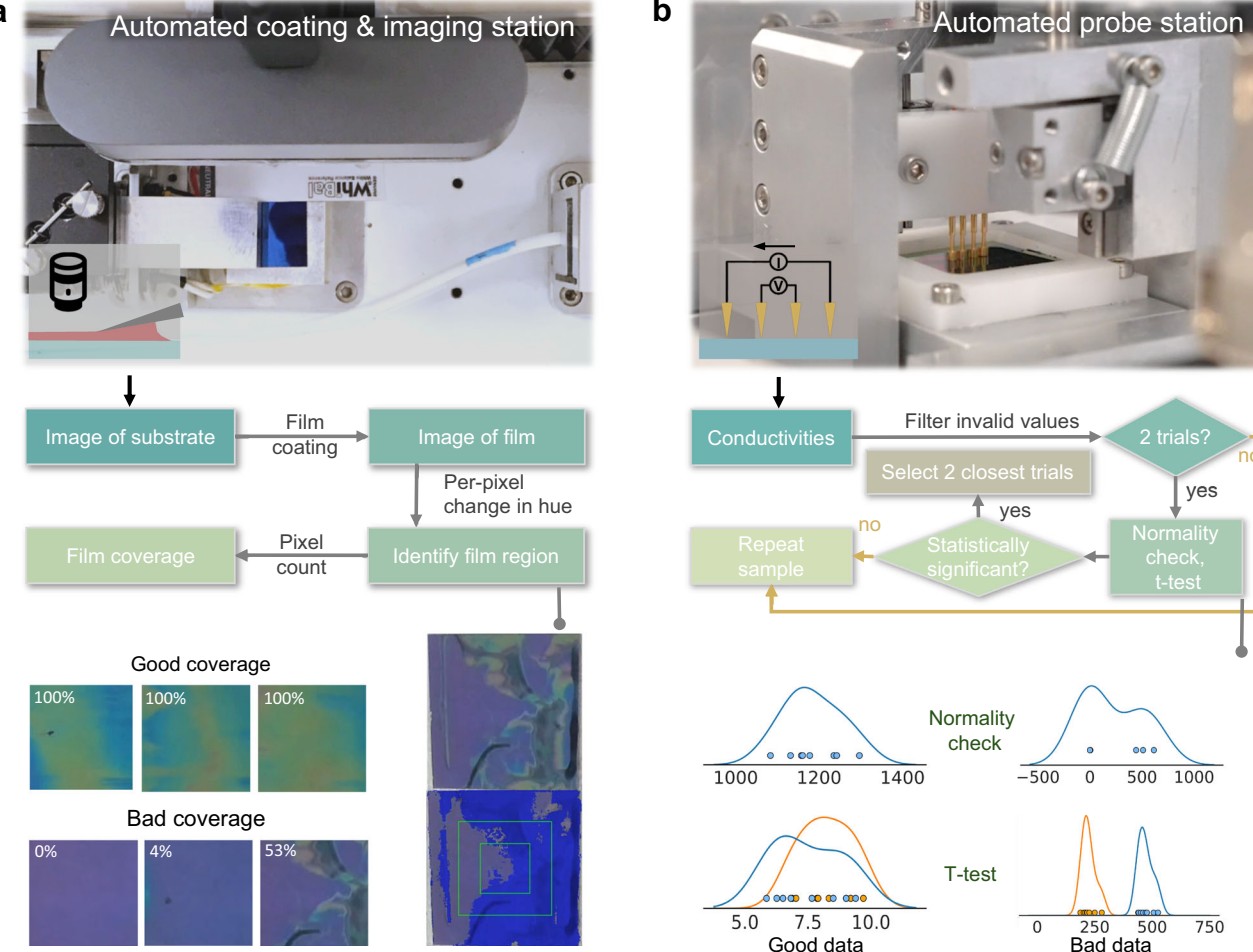

**Fig. 2 | Automated characterization of film defects and electrical conductivity. a** Top-view of the coating and imaging station. Polybot utilizes computer vision to locate the substrate and characterize the thin film sample. A procedure based on color changes is used for the quantification of film coverage percentage. **b** Side- view of the 4-point collinear probe station. Polybot measures current-voltage curves across different regions of the sample and obtain repeatable conductivity values by leveraging a statistical analysis approach.

relatively uniform distribution within the training data (Fig. 3a). From the GPR predicted values while considering data scarcities, Polybot evaluates the expected improvements (EI) in electrical conductivity for all uncharted experimental processing conditions (Supplementary section 2.4). The EI acquisition function balances exploration and exploitation based on a tunable trade-off hyperparameter. Experimental conditions at the top of this EI-ranked list are the most valuable candidates for information gathering or improvements in performance. In a typical BO, the top candidate in this list is always chosen for the next experiment. However, this can be suboptimal due to local minima traps arise from EI overly focusing on the estimated improvements[38]. To alleviate this while considering film coverage as a secondary objective, Polybot employs an importance-guided BO where the list of EI ranked conditions are considered from top to bottom until one condition is selected, and the probability of selecting a particular condition is proportional to the KDE predicted film coverage and clipped to the interval [0.1, 0.9] (Figs. 1c and 3b and Supplementary section 2.5). In this way, Polybot prioritizes improvements in a challenging objective, i.e., film conductivity, while guided by a more achievable objective, i.e., film coverage, which is akin to the concept of importance in probabilistic sampling. Following this iterative learning strategy, Polybot performs the next experiment under the selected processing condition and subsequently refines the prediction models to achieve higher thin film performance using the new data (Fig. 3c).

The progression of our autonomous experiment can be visualized through 2D projections of the 7-dimensional experimental search space, created using the Uniform Manifold Approximation and Projection (UMAP) method (Fig. 3d, e and Supplementary section 2.6)[39]. In the UMAP plot, every experimental condition is depicted as a point, and the distance between two points is proportional to the Euclidean distance between the processing parameter values. The initial training data points (circles) are evenly distributed among all possible conditions (in gray). Polybot, with the use of importance-guided BO, quickly identified regions that maximize both electrical conductivity and film coverage and iteratively improved the sample performance within a small number of samples (triangles). The termination of our experiment is determined based on our initial budget and the achieved thin film performance, i.e., when the experiment exceeds two weeks or when the measured conductivity do not show further improvements after reaching a reasonable expected performance (Supplementary Fig. 5).

From the experimental results, three top-performing experimental processing conditions (Supplementary Table 3) are identified using a Pareto Front analysis (Supplementary section 2.7 and Supplementary Fig. 6). One of these formulation and coating conditions is adapted for large-scale fabrication, as well as subsequent in-depth structural characterizations to understand the changes of PEDOT:PSS in solutions and thin films. First, we demonstrated a successful adaptation of the identified processing conditions on an industrially relevant scale-up blade and

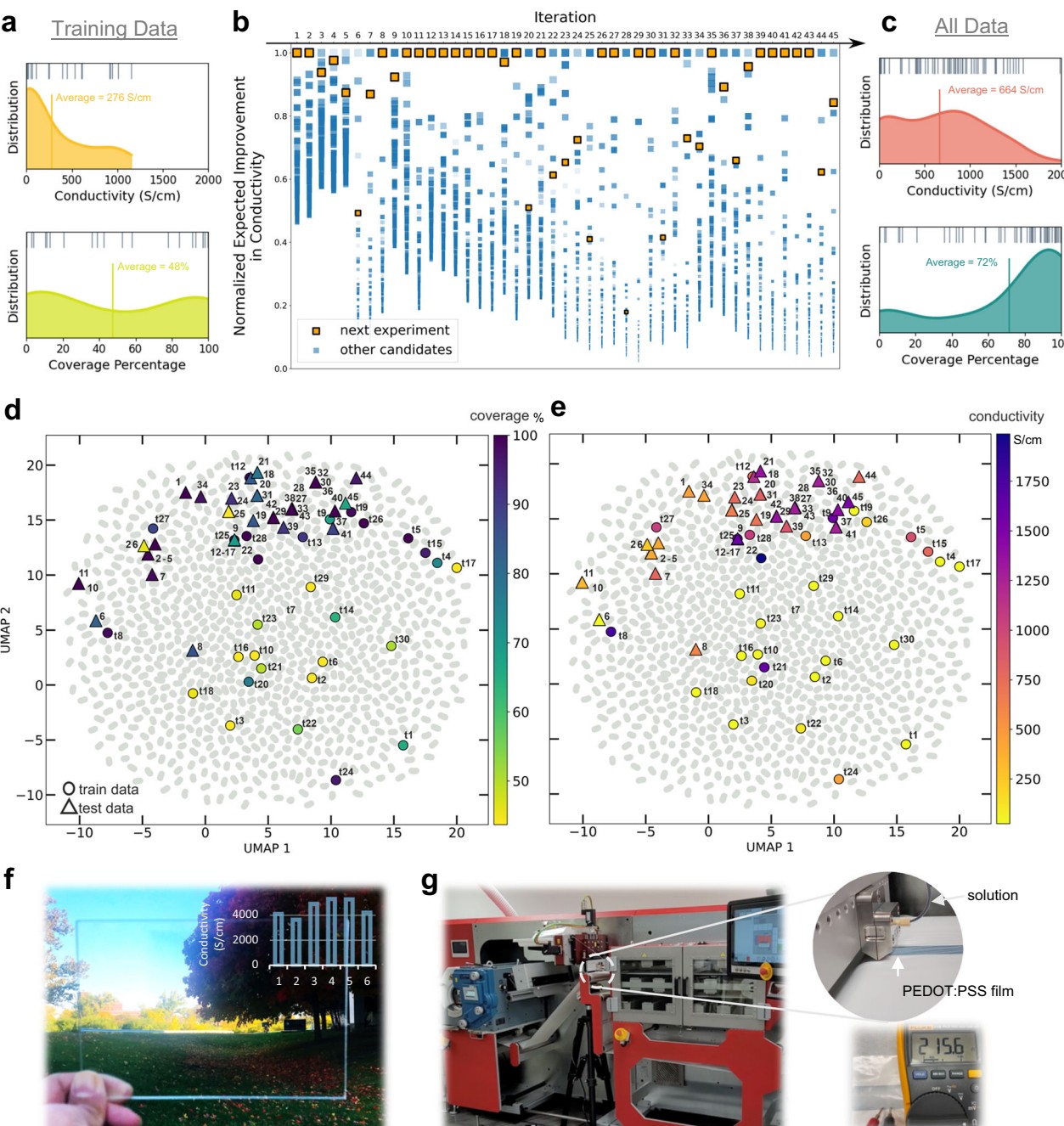

**Fig. 3 | AI-guided closed-loop optimization and their scaled-up fabrication. a–c** Conductivity and coverage distributions before and after the iterative importance-guided BO process. Training data were generated using LHS whereas all data refer to test data proposed by the importance-guided BO algorithm in addition to the initial training data. Each iteration ranks candidates by their expected improvement (EI) in conductivity, sampling based on predicted coverage. For illustration, the top 100 candidates in each experimental iteration are denoted by square markers with size and opacity proportional to their normalized EI in conductivity (by the highest value per iteration) and predicted coverage percentage, respectively. **d, e** Coverage and conductivity evolution illustrated on UMAP-reduced 2D maps of the experimental space. Training data (circle markers, t1–t30) and test data (triangular markers, iterations 1–45) are colored by measured properties. Grey points represent all experimental conditions (933,120 vectors). **f** Blade-coated PEDOT:PSS films with optimized condition, showing conductivity measurements across six locations. **g** Roll-to-roll manufacturing of conductive PEDOT:PSS films on a laminated paper roll.

roll-to-roll coating process (Fig. 3g and Supplementary Movie 2 and 3). Specifically, we have opted for the use of only EG as the additive, at a concentration of 5 vol%, in combination with a relatively low coating speed of 1 mm/s and a relatively high coating temperature of 90 °C. We performed scale-up blade coating of the formulation on a glass substrate that has similar surface chemistry as the SiOx/Si wafer substrates used in the experiments (Fig. 3f). To further remove PSS content and improve PEDOT packing structure, the post-treatment step involving solvent

rinsing was repeated twice more[40]. As a result, the blade-coated 100 cm² film exhibited 100% coverage and achieved an averaged conductivity exceeding 4500 S/cm, placing it among the highest performing PEDOT:PSS films[36,41]. Additionally, we successfully printed a highly conductive film onto a laminated paper roll using a roll-to-roll printer, applying the same conditions (Fig. 3g). To investigate the influence of DMSO on processability in the scale-up manufacturing station, we blade-coated a film from a solution with 2 vol% DMSO, which revealed the

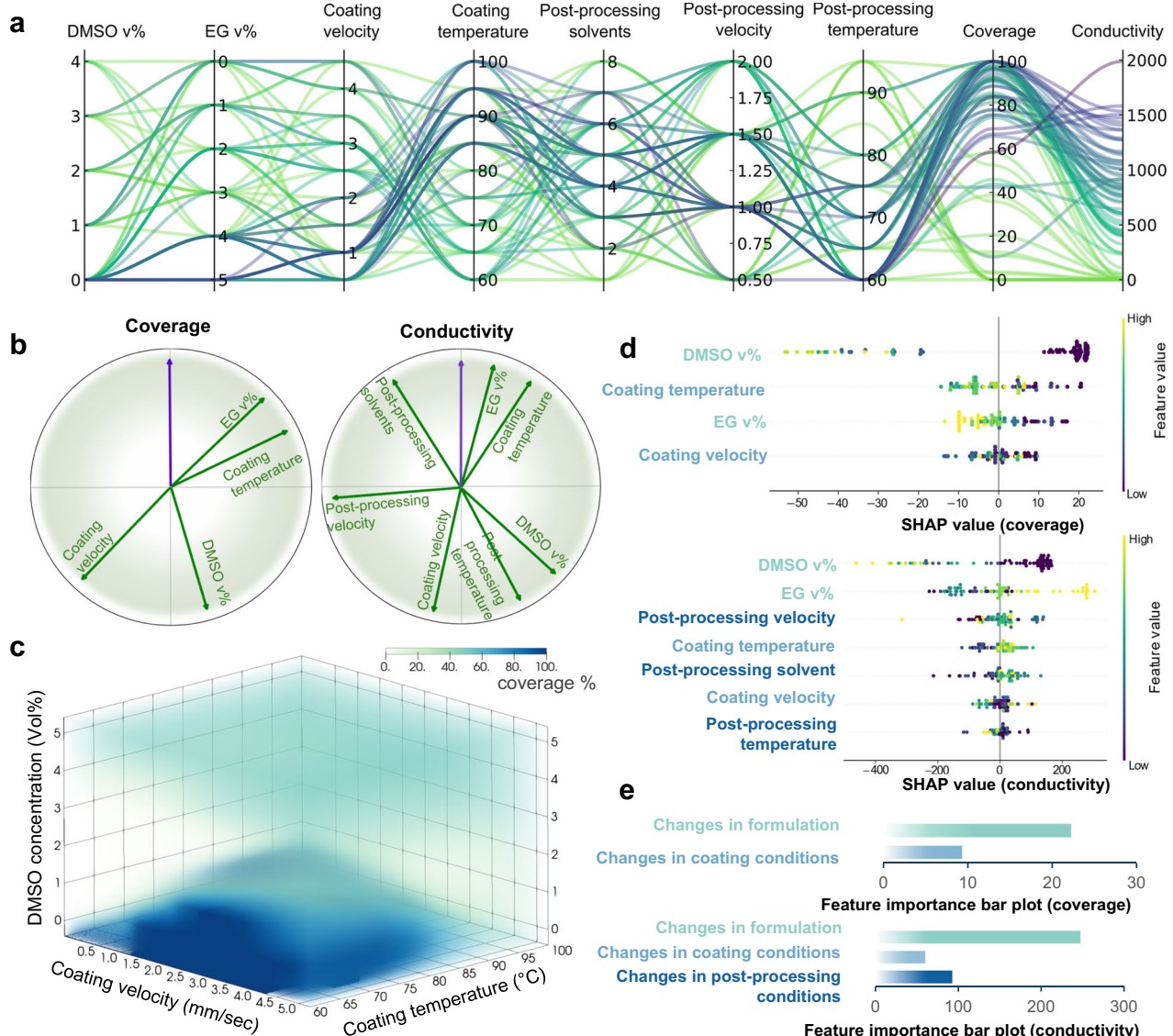

**Fig. 4 | Correlation and interpretability of experimental parameters. a** Parallel coordinates plot showing all the experimental conditions and objectives. The lines are color-coded based on increasing conductivity values (light green indicates the lowest and dark purple the highest conductivity values). **b** Normalized loading plots showing the correlations between the experimental parameters and the thin film properties. **c** Density mapping of the most important parameters that affect the observed film coverage. **d** The feature importance ranking obtained from random forest regression algorithm and SHAP, showing the processing conditions

in descending order. Processing parameters that affect the coverage. (top) The model output in this case is the conductivity of the thin film (bottom). Large positive yellow values increase the conductivity. As a result, the larger the EG concentration, the coating temperature, and the coating velocity the higher the expected conductivity of the film. **e** Bar plots showing the relative influence of condition changes across the three main experimental steps on coverage and conductivity.

formation of numerous defects during the coating process (Supplementary Fig. 10 and Supplementary Movie 4). This observation validates the unfavorable impact of DMSO on processability, consistent with the insights gained through the in-depth analysis (Fig. 4). These findings highlight the successful translation of optimized process parameters from autonomous experiments to scale-up fabrication, paving the way for the production of highly conductive PEDOT:PSS films on large-scale.

### Elucidating design principles from in-depth data analysis and structural characterizations

The data collected from our experiments can be visualized and analyzed to elucidate the principles and optimal conditions for manufacturing low defects, highly conductive PEDOT:PSS thin films. Parallel coordinates plot (Fig. 4a) and normalized loading plots (Fig. 4b) of the data highlight the complex relationships and correlations across the

experimental parameters and thin film performance. In the parallel coordinates plot, each polyline, colored by film conductivity, represents individual experimental conditions and intersects the axes at their corresponding parameter values. In the normalized loading plots (a graphical representation of the correlation matrices shown in Supplementary Fig. 7), the obtuse angle between two arrows represents the inverse cosine of the Pearson correlation coefficient between the respective parameters, i.e., zero correlation when the arrows are perpendicular, positive correlation when the angle is less than 90°, and negative correlation when the angle is greater than 90°. These correlations provide a clear view of the interplay between the thin film properties and experimental parameters. For example, the DMSO concentration strongly and negatively contributes to film coverage, and the averaged conductivity, as shown in the 3D volume density map obtained from the KDE coverage prediction model (Fig. 4c).

SHAP (Shapley Additive exPlanations) feature importance values (Supplementary section 3) are calculated from a ML model that is trained on all data, selected from a suite of different ML models based on their prediction accuracy (Supplementary Tables 6 and 7 and Supplementary Fig. 9). It is noteworthy that in an autonomous experiment driven by BO, the later data points tend to skew towards higher values of the target property (Fig. 3c). Therefore, it is important to implement an efficient binning strategy to partition the data points into bins containing equal quantities. During the training and test process of the ML models, these bins are used for stratified splitting, thereby preserving a normal distribution of the data and improving the model's ability to generalize. Shapley values are used to explain the best performing ML model and provide insights on the parameter importance (Supplementary section 3.3). The summary plots of the SHAP analysis (Fig. 4d) ranked the experimental parameters, from top to bottom along the vertical axis, based on their impact on film coverage and conductivity. Within each parameter, all contributing data points are ordered along the horizontal axis and colored by their values. The results provide insights on the key features contributing to low film defects and high film conductivity. For instance, DMSO concentration is identified as the most influential parameter in achieving highly conductive thin films with low defects. Low DMSO concentration (dark purple) leads to high coverage and conductivity (far right on the horizontal axis) whereas high DMSO concentration (yellow) leads to low coverage and conductivity (far left on the horizontal axis). Overall, the most influential factor for achieving conductive thin films with low defects is the changes in the formulation of the PEDOT:PSS solution, followed by adjustments in processing and post-processing conditions proposed in this research. (Fig. 4e).

It is worth emphasizing that the processability plays a vital role in practical manufacturing, yet it is often challenging to capture accurately in literature. The utilization of autonomous experiments offers a unique opportunity to generate unbiased, systematic and cost-effective data, which, when combined with ML method, helps unveil intricate formulation-processing-property relationships in high-dimensional spaces. Having a quantitative and specific understanding of each factor's influence on the targeted properties enables the design of effective strategies for manufacturing highly conductive PEDOT:PSS films. These findings provide the way for optimizing and tailoring film properties with precision.

To understand the intricate relationship between manufacturing conditions and the enhancement of conductivity, we embarked on an in-depth exploration of both solution-state structures and solid-state morphologies across three representative samples, specifically, one prepared from pristine PEDOT:PSS solution, as well as two others produced under markedly distinct conditions. Cryogenic electron microscopy (cryo-EM) was used to directly visualize the PEDOT:PSS structures in their solution-state (Supplementary Fig. 11), unveiling the emergence of substantial aggregates composed of well-dispersed PEDOT:PSS colloidal particles upon adding small amounts of DMSO and/or EG additives (Fig. 5a). This phenomenon enables greater bridging of the conductive PEDOT phase, ultimately contributing to higher conductivity. From the wide-angle X-ray scattering (GIWAXS) characterization of these three representative samples, the two treated PEDOT:PSS films exhibited a relatively high degree of crystallinity in the PEDOT phase compared to the film spin-coated from pristine solution (Supplementary Fig. 12). Blade-coated films also showed slightly morphological alignments (Fig. 5b and Supplementary Fig. 13), with the champion film, blade-coated from 5 vol% EG additive, exhibiting an interpenetrated fibril network and slightly enhanced vertical phase separation as indicated by a slight increase in the PSS composition on the surface (Supplementary Fig. 14). Together, these morphological features create efficient pathways for charge carriers in PEDOT phase along the coating direction, further enhancing conductivity.

## Discussion
In summary, we demonstrated the synergy between automated laboratory, ML/statistical models, and AI-guided exploration algorithms for the solution manufacturing of highly conductive, low defects polymer thin films. Importance-guided Bayesian optimization was utilized to enable efficient exploration of an intricate 7-dimensional processing space and strategically optimize two key material properties. Our successful experimental campaign led to recipes for scale-up manufacturing of transparent conductive thin films with an average conductivity comparable to the state-of-the-art levels[2,36,41,42]. The conductivity achieved in this work may not be groundbreaking, but the efficiency of our AI-guided robotic processing methodology in identifying the optimal processing pathway is substantial. We envision this AI-assisted automation methodology to not only contribute to the development of smart manufacturing platforms in the field of electronic polymers but also to address the pressing need to enhance system productivity and accelerate materials discovery.

## Methods
### Materials
Poly(3,4-ethylenedioxythiophene) polystyrene sulfonate (PH1000 PEDOT:PSS) was purchased from Heraeus. All solvents, such as dimethyl sulfoxide (DMSO), ethylene glycol (EG), methanol (MeOH), and ethanol (EtOH), were purchased from Sigma-Aldrich and were used as received.

### Stock substrates in the robotic system
4" 300 nm SiO2-covered Si wafers (University Wafer ID 1583) were purchased from University Wafer, Inc., and cut into 2 × 2 cm piece with an automatic wafer dicer (ADT 7122). All the substrates were cleaned by a UV-Ozone cleaner (UVO-Cleaner Model 42 from Jelight Inc.) for 30 min before using. The substrate storage plate can hold 60 substrates a time.

### Stock solutions in the robotic system
DI Water was stocked in 40 ml vial for coating blade cleaning. PEDOT:PSS pristine solution, DMSO, and EG were stocked in 4 ml vials for AI-guided formulation. All post-processing solvents were prepared mixed in ratios listed in Supplementary Table 2 and stocked in 4 ml vials for AI-guided selection.

### Solution preparation
Different amounts of DMSO and EG were automatically added into 1 ml PEDOT:PSS pristine solutions according to the ML suggested formulation. The formed solution was then mixed by a solution heating and mixing module which has a magnetic rotary mechanism that stirred the solution inside the vial with a magnetic stirring bar at a speed of 500 revolutions per minute (rpm) for 60 s. The stirred solution was then placed on the capping and uncapping clamp for pipetting.

### Film coating
The formulated PEDOT:PSS solution was dropped on the substrate and bladed-coated with a custom-built shearing-coater to form a film. The shearing blade consisted of a rectangular piece of silicon, functionalized on the blade surface and blade edge with a monolayer of octadecyltrichlorosilane. The modification of this monolayer enables easy cleaning for the shearing blade after each coating. Films were coated at recommended substrate temperatures and coating speed at a blade height of 50 μm relative to the substrate. After coating is completed, samples were then left on the coating stage for 1 min to dry. If the coating temperature is lower than 70 °C, 2 min of drying time is distributed. The coating speeds and temperatures are determined by ML. As the substrate were dried out, the samples were then transported to

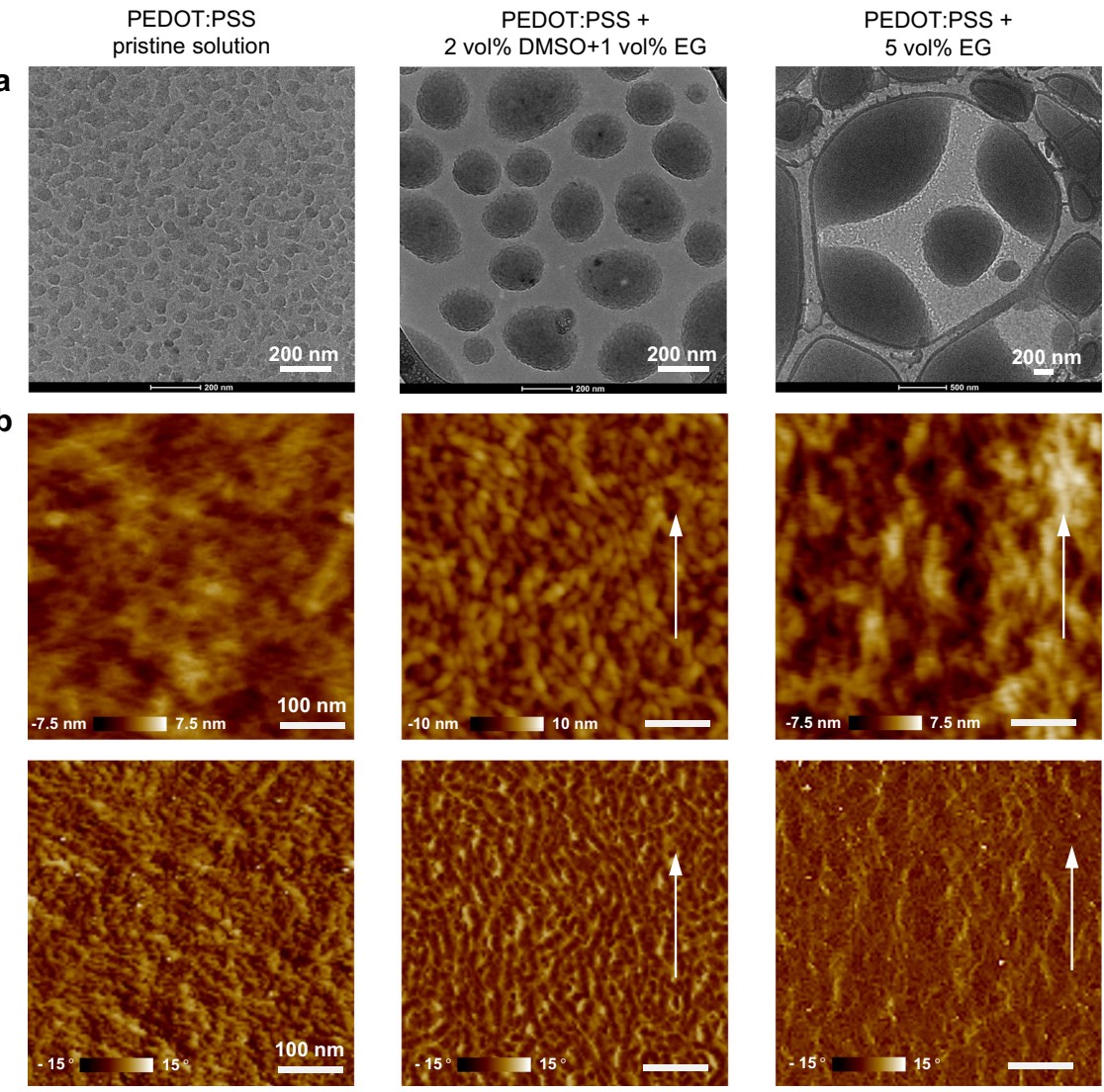

**Fig. 5 | The solution-state structures and solid-state morphologies of PEDOT:PSS.** Cryo-EM (**a**) images of three distinct PEDOT:PSS formulations, and AFM height (**b**, top) and phase (**b**, bottom) images of resulting film samples. Left: Spin-coated film from pristine PEDOT:PSS solution (control). Middle: Blade-coated PEDOT:PSS film from a solution with 2 vol% DMSO and 1 vol% EG, coated at 3 mm/s, 60 °C, and post-processed with methanol/ethanol (1:1), coated at 3 mm/s, 70 °C. Right: Blade-coated PEDOT:PSS film from a solution with 5 vol% EG, coated at 1 mm/s, 90 °C, and post-processed with methanol/ethanol (4:6), coated at 1 mm/s, 60 °C. White arrows indicate blade coating direction. Scale bars in AFM images: 100 nm.

the hot plate at 130 °C for subsequent annealing process. The samples are left on the annealing block for 1 min while the coater blade was cleaned by DI Water. After the annealing is done, the samples were moved back to the coater for post-processing.

### Film post-processing
The PEDOT:PSS films were treated by a post-deposition solvent selected by ML using the solution shearing method. Eight different post-processing solvents with various mixing ratios of MeOH, EtOH and water are listed in Supplementary Table 2. The robotic system would select one solvent for post-treatment based on the ML. Here, the solution-shearing method means wherein solvent was dropped to the front edge of the sample film, then coating blade dragged the solvent across the film. Films were post-processed at a selected temperature and a shearing blade speed recommended by ML. The post-processsing shearing speeds and temperatures variables are also determined by ML. Finally, the film will be left dried out on the coater stage for 30 s and annealed and hot plate (130 °C) for 1 min.

### Thickness characterization
To evaluate the conductivity of the PEDOT:PSS thin films, film thickness needs to be measured. After the robot placed the film on the camera characterization stage with a pneumatic gripper, the film was then characterized by a Filmetrics F40 microscope-based film thickness measurement instrument that outputs the film thickness and goodness of fitting (GOF) data. The data is collected at four different locations (Supplementary Fig. 3) on the thin film. At each location, the F40 would record the thickness data 10 times and choose the thickness data with the largest GOF. After collecting data at all four locations, the data that has GOF lower than 0.9 will be omitted and the measured film thicknesses from the remaining data would be used to calculate the average thicknesses of the left and right sides of the film with left and right two locations. Then, the averaged thicknesses at points on the film (left and right sides) will be used to create a linear model of the thickness of the film through a linear spline interpolation. This model is then used to estimate the thickness of the film at the points where resistance is measured. These estimated thickness values are used in the calculation of resistivity and conductivity. The film thickness of the

large-scale blade-coated film on glass was measured by the step height measurement using the Tencor P-7 stylus profiler.

## Electrical characterization

The conductivity of each PEDOT:PSS film was characterized from eight four-point-probe measurements. Automated probe station and Keithley 4200 were used for this measurement. Keithley Instruments Model 4200A-SCS was used for the electrical characterization measurements on the films. The 4-point collinear probe station is moving across the regions of the film with the sample and produces eight sets of IV measurement data at eight different measuring locations (Supplementary Fig. 3). For each IV measurement at one location on the thin film, the conductivity was then calculated. For a sample of finite width and non-negligible thickness, the resistivity is given by

$$\rho = 4.5324 t \left(\frac{V}{I}\right) f_1 f_2 \qquad (1)$$

here $f_1$ and $f_2$ are correction factors, and $t$ is the film thickness. Since the film thickness is much less than the 4-point probe spacing, $f_1$ is approximately equal to 1. Therefore, the resistivity is given by:

$$\rho \approx 4.5324 t \left(\frac{V}{I}\right) f_2 \qquad (2)$$

where correction factor of $f_2 = 0.925$ is used for the calculation[43].

Subsequently, the conductivity at location $\boldsymbol{i}$ can be calculated as $\sigma i = 1 / \rho_i$, where $i = 1, 2, \ldots 8$. After retrieving the 8 conductivity data points, the data will be processed through interquartile range outlier detection with a scale of 1.5 and average will be calculated. This average conductivity will be used as the final value used by AI. The conductivity of the large-scale blade-coated film on glass was measured by the Filmetrics R50-4PP contact four-point probe system.

## Data availability

Data that support the findings of this study have been deposited in GitHub with the accession name "PEDOT_PSS_supporting_data" (https://github.com/polybot-nexus/PEDOT_PSS_supporting_data/blob/main/PEDOT_experiment.csv). The data are also available within this article and its Supplementary Information.

## Code availability

Source codes and notebooks that demonstrate the interpretation and visualization of data have been deposited in GitHub with the accession name "PEDOT_PSS_supporting_data" (https://github.com/polybot-nexus/PEDOT_PSS_supporting_data).

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

## Acknowledgements

This work was performed at Center for Nanoscale Materials, a U.S. Department of Energy Office of Science User Facility supported by the U.S. DOE, Office of Basic Energy Sciences, under Contract No. DE-AC02-06CH11357. The authors also thank the Materials Engineering Research Facility at Argonne National Lab for providing electronic printing support and National Synchrotron Light Source II at Brookhaven National Laboratory for GIWAXS characterization. J.X., P.D., H.C., Y.W., and A.V. acknowledge the partial support from Laboratory Directed Research and Development (LDRD) funding, provided by the Director, Office of Science, of the U.S. Department of Energy under Contract No. DE-AC02-06CH11357. S.K.R.S.S., H.C., and R.B. acknowledge the supported by the DOE, Office of Science, BES Data, Artificial Intelligence, and Machine Learning at DOE Scientific User Facilities program. J.X. acknowledges the funding support from University of Chicago. We extend our gratitude to our Argonne colleagues and leadership for their support and discussions throughout our collaborations on Polybot.

## Author contributions

J.X. and Y.K. designed the experimental workflow. C.W., H.C., and Y.K. automated the workflow and implemented statistical methods. R.B., H.C., and A.V. integrated the AI/ML algorithms. C.W., Y.K., and H.C. performed the autonomous experiments. A.V., H.C., and R.B. performed the data analysis. A.B. and M.K.Y.C. contributed to the image analysis. L.W., P.D., and S.K.R.S.S. helped in the initial conceptualization of Polybot. Y.L. and H.C.F. performed the Cryo-EM measurement. C.S.M. assisted with the clean room substrate work. N.L. performed the characterization and N.S. performed the GIWAXS experiment. J.X. and H.C. supervised the research. J.X., H.C., and A.V. wrote the manuscript. All the authors contributed to the discussion and manuscript revision.

## Competing interests

The authors declare no competing interests.
