## [Peer Review File · Nature Communications]

REVIEWER COMMENTS

Reviewer #1 (Remarks to the Author):

This paper deals with the optimization of solution formulation, fabrication of thin film coating, and post-treatment of PEDOT:PSS with a Polybot system aimed at improving electrical conductivity. Although the AI-driven automated materials lab is unique and appears to be effective in optimizing better conditions for thin film formation, this paper lacks the polymer science and materials chemistry of PEDOT:PSS necessary to understand the mechanism of conductivity enhancement at the molecular level. Therefore, the reviewer will not recommend publication in Nature Communications, but instead recommend submission to more specialized journals focused on AI-assisted manufacturing.

The PEDOT:PSS is a conductive polymer with a hierarchical structure, so high electrical conductivity can only be achieved by optimizing each structure. The electrical conductivity is determined by carrier density and carrier mobility, and important parameters include the PSS composition ratio, crystallinity, and crystallite distribution.

The effects of secondary dopants such as EG and DMSO, blade-coating condition, and post-treatment on crystallinity, surface morphology, and PSS composition ratio should be clarified by means of XRD, AFM, and XPS.

The thickness of PEDOT:PSS thin films is sensitive to the ambient humidity due to the hygroscopic nature of PSS. How is the atmosphere inside the Polybot system controlled?

PEDOT:PSS with electrical conductivity exceeding 5000 S/cm has already been reported. The authors should compare the obtained results with the literature and clarify the advantages of this paper in terms of hierarchical structure and carrier transport properties.

Reviewer #2 (Remarks to the Author):

In this work, the authors describe an autonomous system for the coverage and conductivity optimization of solution-processed PEDOT-PSS thin films. The developed system couples an automated formulation, processing, and characterization workflow with Bayesian Optimization techniques for selecting new experimental conditions. Processed PEDOT-PSS films were characterized through image processing for film coverage then thickness and 4-point conductivity were used for average film conductivity. Initial training data was created with 30 conditions drawn by Latin Hypercube sampling over the input space. Film conductivity was predicted using a Gaussian process regression model while film coverage was predicted by Gaussian kernel density estimation. Potential experimental candidates were ranked using the Expected Improvement acquisition function on the conductivity then the probability of selecting a candidate was proportional to the predicted film coverage. The new experiment was then performed and the result was used to update the model for the next round of predictions. After the experimental campaign, the top-performing candidates were further characterized for morphology and scale-up. Additionally, the models were investigated through Shapley analysis, parallel coordinate plots, and Normalized loading plots, to determine the influence the input parameters have on the film characteristics. Notably a low DMSO volume fraction was determined to be beneficial in both coverage and conductivity. High-performing films and processing conditions were rapidly identified in a complex input space.

Overall, the developed autonomous lab is very interesting from a technological point of view, and the paper is well-written. The manuscript can be considered for publication after addressing the questions/comments listed below:

1. What was the reasoning for discretizing all of the variables in the design space as opposed to having some continuous and some discrete variables?
2. Do you believe that the step size of the discretization is small enough to identify optima?
3. Were multi-objective optimization techniques considered, such as, randomly weighted Chebyshev scalarized EI (qParEGO) or Pareto Hypervolume Improvement (qNEHVI)
4. As the champion candidates are at the upper limit of the allowed EG range at 5 vol% there may exist an optimum outside the allowed range. Is there a physical limitation on the EG upper bound or just prior experimental conditions? Relaxing arbitrarily imposed formulation constraints could identify superior candidates if the optimization is consistently brushing up against its limits.

Reviewer #3 (Remarks to the Author):

The manuscript by Xu and Chan et al. presents a compelling demonstration of a self-driving lab, named “Polybot” that can efficiently explore a high dimensional processing parameter space to manufacture highly conductive and high coverage thin films of PEDOT:PSS. Polybot is capable of formulating polymer solutions, coat thin films via blade coating, and perform post processing of fabricated films to explore multi-dimensional parameter space to optimize processing conditions that yielded transparent conductive PEDOT:PSS films with averaged conductivity 4500 S/cm. The campaign is driven by a novel importance-guided optimization to simultaneously optimize the primary objective of conductivity by considering the weight of a secondary objective of film coverage. Not stopping there, the authors further unveil intricate formulation-processing-property relationships in high-dimensional spaces. The authors further succeeded in scaling up large-area fabrication of transparent conductors with state-of-the-art performance, while rationalizing the processing-structure-property relationship uncovered. The work represents a major advance in self-driven lab for electronic thin films. The AI component, autonomous experimentation, and in-depth analysis far surpass what was demonstrated in recent work by Berlinguette et al. 10.1126/sciadv.aaz8867. Therefore, the referee recommends publication of this work in Nature Communications if the following minor comments are addressed.

1. The importance of high-throughput experiments has been emphasized throughout the introduction but the actual throughput of Polybot is not mentioned in the manuscript. The authors are encouraged to quantify the efficiency and time-saving aspect of Polybot.
2. In Figure 2b and Supplementary section 3.2, authors demonstrate how the conductivity of PEDOT:PSS films has been conducted. Although the conductivity measurements have been measured at 8 different positions, all of them are perpendicular to the printing direction, and the conductivity parallel to the printing direction might be different as the film has been printed in a single direction, causing the moderate alignment of the polymer which was shown in the last figure. Could the authors comment whether there is anisotropy in conductivity?
3. Could the authors clarify how the range and the stepping of parameters are determined? For instance, the range of velocity seems quite limited.
4. Could the authors comment on the possibility to predict processing conditions through developing predictive models? That would help pointing the way to breaking the record of conductivity, standing at ~8000 S/cm in literature.

Reviewer #4 (Remarks to the Author):

Reviewer #1:

This paper deals with the optimization of solution formulation, fabrication of thin film coating, and post-treatment of PEDOT:PSS with a Polybot system aimed at improving electrical conductivity. Although the AI-driven automated materials lab is unique and appears to be effective in optimizing better conditions for thin film formation, this paper lacks the polymer science and materials chemistry of PEDOT:PSS necessary to understand the mechanism of conductivity enhancement at the molecular level. Therefore, the reviewer will not recommend publication in Nature Communications, but instead recommend submission to more specialized journals focused on AI-assisted manufacturing.

Response: We thank the reviewer for the positive feedback on the effectiveness of our AI-driven automation methodology for guiding electronic thin film processing. We appreciate the reviewer's suggestion to provide more fundamental insights into property enhancement. However, the core innovation of this work lies in the technical merit of developing a research method that autonomously explores complex processing pathways to achieve high-conductivity, low-defect electronic polymer films.

While the structural-property relationships in PEDOT:PSS have been extensively studied, understanding the impact of processing within a 7-dimensional design space—from formulation to post-processing—remains a formidable challenge due to the complex, intertwined relationships involved. Our autonomous processing platform for electronic polymers addresses this challenge by utilizing a robotic laboratory combined with importance-guided Bayesian optimization. This approach allows us to efficiently navigate the 7-dimensional processing space and extract the relative importance of each variable to the overall performance metrics. Based on the knowledge gained from this accelerated research (achieved in just 100 iterations out of a total search space of 933,120), we developed scalable recipes for conductive thin films that achieve an average conductivity exceeding 4500 S/cm. Our work demonstrates the practical effectiveness of combining AI with robotic systems to accelerate research in material processing exploration, offering a model that can inspire a wide range of researchers across various fields. As such, we believe our work appeals to broad audiences and inspires communities beyond single material system.

The PEDOT:PSS is a conductive polymer with a hierarchical structure, so high electrical conductivity can only be achieved by optimizing each structure. The electrical conductivity is determined by carrier density and carrier mobility, and important parameters include the PSS composition ratio, crystallinity, and crystallite distribution. The effects of secondary dopants such as EG and DMSO, blade-coating condition, and post-treatment on crystallinity, surface morphology, and PSS composition ratio should be clarified by means of XRD, AFM, and XPS.

Response: We agree with the reviewer that comprehensive characterizations are essential for understanding the various effects and structure-processing-property relationships of PEDOT:PSS. In fact, many studies have extensively examined the structural-property relationships in PEDOT as briefly summarized as below. For instance, the dopants EG and DMSO have been reported to enhance the relative degree of crystallinity in PEDOT¹, more chain-like nanostructures at the surface are observed after the addition of DMSO and EG, compared to the agglomeration of oblate ellipsoidal-shaped PEDOT nanocrystals observed in undoped PEDOT:PSS (Fig. R1a)². And a slightly increase in the ratio of PEDOT to PSS, as estimated from the XPS S2p spectra (Fig. R1b)². The blade-coating method has been reported to slightly promote the growth of prominent fibers, compared to the spherical particles observed in spin-coated films (Fig. R1c), enhancing vertical phase separation with more PSS on the surface (Fig. R1d), and subtly align the conductive PEDOT species along the shearing direction, as indicated by polarized UV-vis absorption(Fig. R1e)³. Alcohol post-treatment on printed PEDOT has shown a slight increased roughness of the surface morphology of PEDOT before and after solvent post-treatment, and a subtle increase in the PSS/PEDOT component ratio (Fig. R1f, g)⁴. Additionally, there is a higher degree of crystalline structure in PEDOT after MeOH solvent treatment (Fig. R1h)⁵.

Figure R1. The brief summary of previous studies on the individual variables in our work effect on PEDOT:PSS morphology. (a-b), The influence of EG and DMSO on the PEDOT:PSS film morphology. (a). AFM height images of PEDOT:PSS films prepared from pristine solution and solution with the addition of 3% DMSO and 7% EG, (b) S 2p XPS spectra of PEDOT:PSS films prepared from pristine solution and with the addition of 3% DMSO and 7% EG. (c-e), The effect of blade coating conditions on PEDOT:PSS morphology. (c) AFM height images of PEDOT:PSS films prepared from spin casted methods and solution shearing method at shearing speeds of 0.5 mm/s and 3 mm/s, (d) High-resolution XPS C₆₀ ion gun sputtering profiles of solution sheared PEDOT:PSS at 3 mm/s, (e) Polarized UV-vis absorption of solution sheared PEDOT:PSS film at 3 mm/s. (f-h) Morphology of the PEDOT:PSS films before and after solvent post-treatment. (f) AFM height images of PEDOT:PSS films before

and after MeOH post-treatment. (g) S (2p) XPS spectra of untreated and alcohols-treated PEDOT:PSS films. (h) x-ray diffraction patterns of pristine and solvent-treated PEDOT:PSS thin films.

In contrast to previous systematic morphological characterizations, this work focuses on performing in-depth analysis of the solution-state structures of selected representative samples, guided primarily by statistical data analysis. Specifically, we found that changes in formulation influenced conductivity more significantly than variations in coating speed or post-treatment conditions within the entire search space in this project as revealed by Shapley analysis and feature importance bar plot (Fig. 4).

To gain deeper insights into the solution-state structures of PEDOT:PSS after introducing EG and DMSO additives, we employed Cryo-EM technology, which revealed a dramatic transformation in the solution-state structure, even with trace amounts of additives. This solution-structural shift led to the formation of more fibril-like structures in the resulting solid films, as confirmed by AFM characterizations. To further enhance our understanding of the relationship between morphological structure and conductivity, we further conducted grazing wide-angle X-ray scattering (GIWAXS) and XPS characterizations on the three representative samples. GIWAXS characterization showed that the high-conductivity PEDOT:PSS films exhibited a relatively high degree of crystallinity in the PEDOT phase (Fig. R2a, new Fig. S12). XPS S (2p) characterization showed a slight increase in the PSS composition on the surface of the high-conductivity blade-coated PEDOT:PSS film compared to the control spin-coated film, which could suggest potential vertical phase separation in the blade-coated sample (Fig. R2b, new Fig. S14). These findings are generally consistent with previously published studies.

Figure R2. (a) GIWAXS 2D images of three PEDOT:PSS thin films. Left: Spin-coated film from pristine PEDOT:PSS solution (control). Middle: Blade-coated PEDOT:PSS film from a solution with 2 vol% DMSO and 1 vol% EG, coated at 3 mm/s, 60°C, and post-processed with methanol/ethanol (1:1), coated at 3 mm/s, 70°C. Right: Blade-coated PEDOT:PSS film from a solution with 5 vol% EG, coated at 1 mm/s, 90°C, and post-

processed with methanol/ethanol (4:6), coated at 1 mm/s, 60°C. GIWAXS measurements were performed at beamline 11-BM of National Synchrotron Light Source II, Brookhaven National Laboratory. The samples were tilted at incident angle of 0.12° with respect to incident beam. All images were collected under vacuum with an incident beam energy of 13.5 keV and calibrated with silver behenate. For images taken, the area detector was translated vertically for a second exposure. The two images were combined to eliminate gaps due to rows of inactive pixels at the borders between modules. **(b)** S (2p) XPS spectra of spin-coated film from pristine PEDOT:PSS solution (left) and blade-coated PEDOT:PSS film from a solution with 5 vol% EG, coated at 1 mm/s, 90°C, and post-processed with methanol/ethanol (4:6), coated at 1 mm/s, 60°C. The respective band between 166 and 171 eV is assigned to the sulfur atom in PSS, and the doublet peaks between 162 and 166 eV correspond to the sulfur atom in the PEDOT benzene ring. The X-ray photoelectron spectroscopy (XPS) was done with Kratos AXIS Nova with a monochromatic Al K α X-ray source and a delay line detector (DLD) system.

In the revised manuscript, we have added the following text, highlighted in yellow, on Page 18 to further discuss these structure-property relationships: “*From the wide-angle X-ray scattering (GIWAXS) characterization of these three representative samples, the two treated PEDOT:PSS films exhibited a relatively high degree of crystallinity in the PEDOT phase compared to the film spin-coated from pristine solution (Fig. S12). Blade-coated films also showed slightly morphological alignments (Fig. 5b, Fig. S13), with the champion film, blade-coated from 5 vol% EG additive, exhibiting an interpenetrated fibril network and slightly enhanced vertical phase separation as indicated by a slight increase in the PSS composition on the surface (Fig. S14). Together, these morphological features create efficient pathways for charge carriers in PEDOT phase along the coating direction, further enhancing conductivity.*”

The thickness of PEDOT:PSS thin films is sensitive to the ambient humidity due to the hygroscopic nature of PSS. How is the atmosphere inside the Polybot system controlled

Response: We agree with the reviewer that the thickness of PEDOT thin films is indeed sensitive to ambient humidity. To address this, our experiments were conducted under an inert atmosphere, created by nitrogen (N₂) purging, within a sealed enclosure.

For clarification, we have added the following sentence to the revised Supporting Information (Section 3.2 Automated Thickness Measurements), highlighted in yellow:

“Due to the hygroscopic nature of PSS, the thickness of PEDOT:PSS thin films is sensitive to ambient humidity. To ensure an inert atmosphere, our robotic system is enclosed and purged with nitrogen (N₂).”

PEDOT:PSS with electrical conductivity exceeding 5000 S/cm has already been reported. The authors should compare the obtained results with the literature and clarify the advantages of this paper in terms of hierarchical structure and carrier transport properties.

Response: We agree with the reviewer that the electrical conductivity achieved in this work is not groundbreaking. However, the efficiency of our AI-guided robotic processing methodology in

identifying the optimal processing pathway is quite impressive. Considering the search space of 933,120 possibilities, we identified a recipe for achieving 4500 S/cm films in just 80 trials (~30 minutes each, totalling less than 48 hours). Furthermore, obtaining PEDOT with electrical conductivity exceeding 4500 S/cm in large-scale printed films (>10 cm) is rarely reported and remains challenging, even though values exceeding 4500 S/cm have been achieved in smaller sample sizes (1~2 cm scale).

As discussed, our champion film shows a slightly aligned interpenetrated fibril network with relatively high degree of crystallinity in the PEDOT phase as well as a bit promoted vertical phase separation, which are consistent with the morphological features previously observed in highly conductive PEDOT films. However, what sets this work apart is the detailed insight we provide into the relative importance of condition changes during each processing step. These insights, which were challenging to quantitatively extract in the past, are valuable for researchers focusing on condition optimization. For instance, we observed that changes in formulation had the most significant influence on conductivity compared to changes in the post-treatment solvent combination. Therefore, during the optimization process, we prioritize adjustments to the formulation conditions over post-treatment solvent formulation changes.

To clarify this, we have added the following sentence highlighted in yellow in the Conclusions:

“The conductivity achieved in this work may not be groundbreaking, but the efficiency of our AI-guided robotic processing methodology in identifying the optimal processing pathway is substantial.”

Reviewer #2:

In this work, the authors describe an autonomous system for the coverage and conductivity optimization of solution-processed PEDOT-PSS thin films. The developed system couples an automated formulation, processing, and characterization workflow with Bayesian Optimization techniques for selecting new experimental conditions. Processed PEDOT-PSS films were characterized through image processing for film coverage then thickness and 4-point conductivity were used for average film conductivity. Initial training data was created with 30 conditions drawn by Latin Hypercube sampling over the input space. Film conductivity was predicted using a Gaussian process regression model while film coverage was predicted by Gaussian kernel density estimation. Potential experimental candidates were ranked using the Expected Improvement acquisition function on the conductivity then the probability of selecting a candidate was proportional to the predicted film coverage. The new experiment was then performed and the result was used to update the model for the next round of predictions. After the experimental campaign, the top-performing candidates were further characterized for morphology and scale-up. Additionally, the models were investigated through Shapley analysis, parallel coordinate plots, and Normalized loading plots, to determine the influence the input parameters have on the film

characteristics. Notably a low DMSO volume fraction was determined to be beneficial in both coverage and conductivity. High-performing films and processing conditions were rapidly identified in a complex input space. Overall, the developed autonomous lab is very interesting from a technological point of view, and the paper is well-written. The manuscript can be considered for publication after addressing the questions/comments listed below:

Response: We thank the reviewer for the positive feedback on this work.

1. What was the reasoning for discretizing all of the variables in the design space as opposed to having some continuous and some discrete variables?

Response: The discretization of variables was implemented to capture the significant precision of our experimental parameters. Specifically, it ensures that the smallest change in each parameter is meaningful, considering their sensitivity to the experiment and the tunability of our hardware modules. This approach helps narrow the design space and allows the search algorithms to more efficiently explore the entire space.

2. Do you believe that the step size of the discretization is small enough to identify optima?

Response: Thank you for this critical question. The selection of the step size was a careful balance between capturing the optima as effectively as possible within our hardware's capabilities and avoiding unnecessary expansion of the search space. The step size for formulation (50 μ L/1vol% additive increments) and temperature (5°C increments) was based on the reliable resolution of the hardware modules. For the coating conditions, we chose a step size of 0.5 mm/s based on previous literatures^{3, 6} and our initial in-house sensitivity tests. The optima we identified are within our design space, which includes a total of 933,120 possible experimental conditions. While it is possible to further refine the optima by reducing the step size, this is not strictly necessary for rapid screening. The main goal of this AI-guided robotic optimization is to identify the region that should be the focus for future scaling up the process.

3. Were multi-objective optimization techniques considered, such as, randomly weighted Chebyshev scalarized EI (qParEGO) or Pareto Hypervolume Improvement (qNEHVI).

Response: Yes, we have carefully thought about the suitable optimization techniques prior to setting up of our autonomous PEDOT : PSS experiments. Given the complexity of our experimental procedures and the nature of working with real data in an automatic/iterative manner, we could not perform a fair comparison between different optimization techniques during the experiments and resort to choosing

the most promising technique based on our expectations and notable characteristics of the experiments, such as 1) small data due to time-consuming experimental procedures, 2) noise/uncertainty of the measurement intrinsic to the materials system, and 3) optimization of both electrical conductivity and film uniformity. Bayesian optimization with distributions modelled using Gaussian Processes was selected as the sample efficient probabilistic algorithm for addressing point 1 and 2, while a specially tailored multi-objective scheme was utilized to handle point 3 and 2. Prior to the selection of multi-objective techniques, we analysed 30 uniformly sampled samples (the training dataset) and realized that unlike our initial expectation, electrical conductivity and film uniformity of PEDOT-PSS thin films do not appear to be objectives involving trade-offs and there is clear hierarchy between them, i.e., achieving one helps achieving the other and film uniformity is the easier, more achievable goal. Our importance guided multi-objective approach is designed specifically to leverage these aspects of the experiments with simple implementations. Typical multi-objective approaches like the ones mentioned by the reviewer have advantages when handling trade-offs (e.g., pareto front). For instance, qParEGO utilizes scalarization of weights to effectively treat multiple objectives as a single objective while qNEHVI focuses on improving hypervolume for locating a set of pareto-optimal solutions under the consideration of data noise. Though these approaches can still be beneficial in our case, they are likely less relevant or effective for non-trade off objectives. Furthermore, our autonomous experiments are serial and sequential, so there is no parallel processing of sample batches that can take advantage of a set of pareto-optimal solutions/suggestions. The result from the autonomous run validates our observations and the effectiveness of the selected multi-objective optimization approach. In future studies, we would consider evaluating the use of algorithms like qParEGO and qNEHVI in autonomous experiments for materials systems with property trade-offs.

4. As the champion candidates are at the upper limit of the allowed EG range at 5 vol% there may exist an optimum outside the allowed range. Is there a physical limitation on the EG upper bound or just prior experimental conditions? Relaxing arbitrarily imposed formulation constraints could identify superior candidates if the optimization is consistently brushing up against its limits.

Response: Thank you for this great question. The upper limit of ethylene glycol (EG) concentration in this design space was primarily determined by previous experimental conditions and the consideration of drying time. From prior literature on optimizing EG-doped PEDOT films, it has been observed that a range of 3-5 vol% EG generally provides good overall performance for PEDOT⁷⁻⁹. Another consideration is based on our in-house preliminary testing. The low vapor pressure of EG can result in very slow drying speeds, especially when applied at lower temperatures, which can lead to uneven drying. This lack of uniformity can produce inconsistent and non-reproducible data, particularly in such rapid screening experiments using the Polybot system. To ensure consistent results, we found that

keeping the concentration below 5 vol% was preferred. Thus, we set the 5 vol% as the upper limit for this variable in this rapid optimization platform.

But, we acknowledged this problem too, and enhanced our decision-making workflow by incorporating more real-time features, such as data visualization and feature importance analysis, in our next autonomous material discovery project, which will be submitted in near future. These tools aim to address the problem. For instance, if a critical feature is identified and consistently pushes boundary limits, the system can prompt human operators to make appropriate adjustments accordingly.

Reviewer #3:

The manuscript by Xu and Chan et al. presents a compelling demonstration of a self-driving lab, named “Polybot” that can efficiently explore a high dimensional processing parameter space to manufacture highly conductive and high coverage thin films of PEDOT:PSS. Polybot is capable of formulating polymer solutions, coat thin films via blade coating, and perform post processing of fabricated films to explore multi-dimensional parameter space to optimize processing conditions that yielded transparent conductive PEDOT:PSS films with averaged conductivity 4500 S/cm. The campaign is driven by a novel importance-guided optimization to simultaneously optimize the primary objective of conductivity by considering the weight of a secondary objective of film coverage. Not stopping there, the authors further unveil intricate formulation-processing-property relationships in high-dimensional spaces. The authors further succeeded in scaling up large-area fabrication of transparent conductors with state-of-the-art performance, while rationalizing the processing-structure-property relationship uncovered. The work represents a major advance in self-driven lab for electronic thin films. The AI component, autonomous experimentation, and in-depth analysis far surpass what was demonstrated in recent work by Berlinguette et al. 10.1126/sciadv.aaz8867. Therefore, the referee recommends publication of this work in Nature Communications if the following minor comments are addressed.

Response: We appreciate the reviewer’s positive comment!

1. The importance of high-throughput experiments has been emphasized throughout the introduction but the actual throughput of Polybot is not mentioned in the manuscript. The authors are encouraged to quantify the efficiency and time-saving aspect of Polybot.

Response: Thanks for this question. Polybot can complete an entire experimental loop—formulation, processing, post-processing, and conductivity measurement—in approximately 15 minutes per sample, allowing for a throughput of around 100 samples per day. In comparison, based on our in-house testing, human researchers typically produce around 10–20 samples per day, due to the dispersed location of

instruments and the manual nature of the work. Additionally, Polybot demonstrates much greater repeatability in experimental results compared to human scientists.

We have added the following sentence to talk about the throughput to the manuscript at the end of the first paragraph highlighted in yellow in the Results section: “*The automated platform can complete an entire experimental loop—formulation, processing, post-processing, and conductivity measurement—in approximately 15 minutes per sample, enabling a throughput of around 100 samples per day with great repeatability.*”

2. In Figure 2b and Supplementary section 3.2, authors demonstrate how the conductivity of PEDOT:PSS films has been conducted. Although the conductivity measurements have been measured at 8 different positions, all of them are perpendicular to the printing direction, and the conductivity parallel to the printing direction might be different as the film has been printed in a single direction, causing the moderate alignment of the polymer which was shown in the last figure. Could the authors comment whether there is anisotropy in conductivity?

Response: There is slight anisotropy in conductivity as measured on the champion film (EG 5%) in the last figure, with the ratio of conductivities measured parallel and perpendicular to the printing direction being around 1.2. We also performed polarized UV-Vis measurements on this champion film (Fig. R3), which revealed slight anisotropy in morphology, consistent with the moderate alignment of the polymer, as shown in the AFM image in the last figure. However, the anisotropy phenomenon induced by directional coating is not as pronounced in PEDOT:PSS films as in other conjugated polymer systems (e.g., DPP-based, P3HT). Additionally, the feature importance analysis also reveals that the changes in coating conditions within this design space had the least significant impact on improving conductivity. We think this is likely due to the PEDOT:PSS’s colloid-like solution-state structure.

To clarify this, we have added the following sentence to the last paragraph of the Results Section and added Fig. R3 as Fig. S13 in the manuscript: “*Blade-coated films also showed slightly morphological alignments (Fig. 5b, Fig. S13), Together, these morphological features create efficient pathways for charge carriers in the PEDOT phase along the coating direction, further enhancing conductivity.*”

Figure R3. Polarized ultraviolet–visible absorption spectra of blade-coated PEDOT:PSS film from a solution with 5 vol% EG, coated at 1 mm/s, 90°C, and post-processed with methanol/ethanol (4:6), coated at 1 mm/s, 60°C. The red (Para) curve was obtained when the film's coating direction was aligned with the polarizer axis, whereas the gray curve (Perp) was collected with the coating direction perpendicular to the polarizer axis.

3. Could the authors clarify how the range and the stepping of parameters are determined? For instance, the range of velocity seems quite limited.

Response: The selection of the range and stepping size was determined based on prior experimental knowledge established in relevant PEDOT literature, as well as a careful balance between capturing the optima effectively within our hardware's capabilities and avoiding unnecessary expansion of the search space.

For additives, the literature indicates that a range of 3-5 vol% generally provides optimal performance for PEDOT:PSS⁷⁻⁹. We chose a formulation step size of 50 μ L (1 vol% increments) based on the resolution capabilities of our liquid handling system. Similarly, for coating temperatures, the range of 60-100°C was determined based on the boiling points of water and the solvent, while also balancing drying speed. A 5°C increment was selected to achieve reliable resolution from our heated stage.

The coating speed range, from 0.5 to 5 mm/s, was determined by both previous research (refs, around 0.25-5 mm/sec) and in-house preliminary experiments. Our pre-test revealed that at speeds below 0.5 mm/s, the film often exhibited poor uniformity. At speeds above 5 mm/s, the deposition already entered the Landau-Levich region, where a liquid layer is pulled out from the blade and dried later. The stepping size was set at 0.5 mm/s, based on literature and to avoid unnecessary expansion of the search space.

The boundary of the design space and the stepping size in this work are rigid. We also recognized that improving the human-machine interface is crucial for relaxing formulation constraints and potentially identifying superior candidates, especially when optimization approaches its boundaries. In our current self-driving laboratory, we have enhanced our decision-making workflow with real-time features such as data visualization, trendline monitoring, and feature importance analysis. These tools allow for better real-time insights and human intervention when necessary. For instance, if a particular feature

consistently pushes against boundary/stepping limits, the system can prompt human operators to make realistic adjustments accordingly.

4. Could the authors comment on the possibility to predict processing conditions through developing predictive models? That would help pointing the way to breaking the record of conductivity, standing at ~8000 S/cm in literature.

Response: Thank you for this constructive question. Currently, due to the relatively small dataset generated from the autonomous experiment (75 data points), developing a robust inverse design model presents challenges at this stage. However, we think that achieving this could be possible through the following developments: 1. expanding the dataset with more design variables and finer step sizes to capture more nuanced relationships; 2. implementing better model selection strategies to identify the most accurate predictive models through real-time performance comparisons; 3. enhancing the human-machine interface, allowing human intervention for adjustments and incorporating new hypotheses to refine the predictive models. This will be a continuous project for us to explore further.

Reviewer #4:

Response: We thank the reviewer for carefully reading our work and providing constructive feedback.

Reference:

1. Yousefian, H.; Babaei-Ghazvini, A.; Isari, A. A.; Hashemi, S. A.; Acharya, B.; Ghaffarkhah, A.; Arjmand, M., Solvent-doped PEDOT:PSS: Structural transformations towards enhanced electrical conductivity and transferable electromagnetic shields. *Surfaces and Interfaces* **2024**, *51*, 104481.
2. Thomas, J. P.; Zhao, L.; McGillivray, D.; Leung, K. T., High-efficiency hybrid solar cells by nanostructural modification in PEDOT:PSS with co-solvent addition. *Journal of Materials Chemistry A* **2014**, *2* (7), 2383-2389.
3. Worfolk, B. J.; Andrews, S. C.; Park, S.; Reinspach, J.; Liu, N.; Toney, M. F.; Mannsfeld, S. C. B.; Bao, Z., Ultrahigh electrical conductivity in solution-sheared polymeric transparent films. *Proceedings of the National Academy of Sciences* **2015**, *112* (46), 14138-14143.
4. Li, Q.; Yang, J.; Chen, S.; Zou, J.; Xie, W.; Zeng, X., Highly Conductive PEDOT:PSS Transparent Hole Transporting Layer with Solvent Treatment for High Performance Silicon/Organic Hybrid Solar Cells. *Nanoscale Research Letters* **2017**, *12* (1), 506.

5. Wang, X.; Kyaw, A. K. K.; Yin, C.; Wang, F.; Zhu, Q.; Tang, T.; Yee, P. I.; Xu, J., Enhancement of thermoelectric performance of PEDOT:PSS films by post-treatment with a superacid. *RSC Advances* **2018**, *8* (33), 18334-18340.
6. Wang, G.; Huang, W.; Eastham, N. D.; Fabiano, S.; Manley, E. F.; Zeng, L.; Wang, B.; Zhang, X.; Chen, Z.; Li, R.; Chang, R. P. H.; Chen, L. X.; Bedzyk, M. J.; Melkonyan, F. S.; Facchetti, A.; Marks, T. J., Aggregation control in natural brush-printed conjugated polymer films and implications for enhancing charge transport. *Proc Natl Acad Sci U S A* **2017**, *114* (47), E10066-e10073.
7. Dimitriev, O. P.; Grinko, D. A.; Noskov, Y. V.; Ogurtsov, N. A.; Pud, A. A., PEDOT:PSS films—Effect of organic solvent additives and annealing on the film conductivity. *Synthetic Metals* **2009**, *159* (21), 2237-2239.
8. Fan, Z.; Ouyang, J., Thermoelectric Properties of PEDOT:PSS. *Advanced Electronic Materials* **2019**, *5* (11), 1800769.
9. Jeong, H. J.; Jang, H.; Kim, T.; Earmme, T.; Kim, F. S., Sigmoidal Dependence of Electrical Conductivity of Thin PEDOT:PSS Films on Concentration of Linear Glycols as a Processing Additive. *Materials* **2021**, *14* (8), 1975.

REVIEWERS' COMMENTS

Reviewer #1 (Remarks to the Author):

The authors have revised the manuscript incorporating the reviewers' comments and advice. Therefore, the reviewer recommends publication in this form.

Reviewer #2 (Remarks to the Author):

The authors have properly addressed the comments and questions raised by the reviewer. The manuscript can be recommended for publication in its current form.

Reviewer #3 (Remarks to the Author):

I appreciate that the authors have made a great effort to address our concerns. Their additional experiment, and explanation helped to significantly improve the quality of the manuscript. There are two minor suggestions which the authors could choose to consider further. But the manuscript can be accepted in its current form.

1. The authors selected solvent formulation, coating speed, coating temperature, post-processing solvent/temperature/speed as the 7 parameters determining the processing space they searched through. It seems the authors fixed the blade height(50um) and the volume(3.5uL) of the precursor solution when printing, which might affect the curvature and size of the meniscus and therefore the coating outcome. These parameters could be adjusted in future works as it affects the back meniscus.

2. It is interesting that the authors were able to identify a recipe to achieve 4500 S/cm in just less than 80 trials out of 933,120 possibilities, and that low DMSO volume fraction was beneficial to both coverage and conductivity, based on less than 0.01% (75 datapoints) of total parameter space. For future work, they could consider to go a step further to develop a physical-based hypothesis and further experiments to validate the hypothesis.

Reviewer #4 (Remarks to the Author):

REVIEWER

COMMENTS

Reviewer #1:

The authors have revised the manuscript incorporating the reviewers' comments and advice. Therefore, the reviewer recommends publication in this form.

Response: We thank the reviewer for all the suggestions.

Reviewer #2:

The authors have properly addressed the comments and questions raised by the reviewer. The manuscript can be recommended for publication in its current form.

Response: We thank the reviewer for their positive feedback on this work.

Reviewer #3:

I appreciate that the authors have made a great effort to address our concerns. Their additional experiment, and explanation helped to significantly improve the quality of the manuscript. There are two minor suggestions which the authors could choose to consider further. But the manuscript can be accepted in its current form.

- 1. The authors selected solvent formulation, coating speed, coating temperature, post-processing solvent/temperature/speed as the 7 parameters determining the processing space they searched through. It seems the authors fixed the blade height(50um) and the volume(3.5uL) of the precursor solution when printing, which might affect the curvature and size of the meniscus and therefore the coating outcome. These parameters could be adjusted in future works as it affects the back meniscus.*
- 2. It is interesting that the authors were able to identify a recipe to achieve 4500 S/cm in just less than 80 trials out of 933,120 possibilities, and that low DMSO volume fraction was beneficial to both coverage and conductivity, based on less than 0.01% (75 datapoints) of total parameter space. For future work, they could consider to go a step further to develop a physical-based hypothesis and further experiments to validate the hypothesis.*

Response: We thank the reviewer for the suggestions for our future work.

- 1. The authors selected solvent formulation, coating speed, coating temperature, post-processing solvent/temperature/speed as the 7 parameters determining the processing space they searched through. It seems the authors fixed the blade height(50um) and the volume(3.5uL) of the precursor solution when*

printing, which might affect the curvature and size of the meniscus and therefore the coating outcome. These parameters could be adjusted in future works as it affects the back meniscus.

Response: We thank the reviewer for the great suggestions. We will test these two variables in our future work.

2. It is interesting that the authors were able to identify a recipe to achieve 4500 S/cm in just less than 80 trials out of 933,120 possibilities, and that low DMSO volume fraction was beneficial to both coverage and conductivity, based on less than 0.01% (75 datapoints) of total parameter space. For future work, they could consider to go a step further to develop a physical-based hypothesis and further experiments to validate the hypothesis.

Response: We thank the reviewer for the constructive suggestions.